# The Potential Benefic Effect of Nicotinamide Riboside in Treating a Murine Model of Monoiodoacetate-Induced Knee Osteoarthritis

**DOI:** 10.3390/jcm12216920

**Published:** 2023-11-03

**Authors:** Florin-Liviu Gherghina, Ion Mîndrilă, Sandra-Alice Buteică, George Bică, Cătălina-Gabriela Pisoschi, Cristina-Elena Biță, Iulia-Alexandra Paliu, Otilia-Constantina Rogoveanu

**Affiliations:** 1Department of Physical Medicine and Rehabilitation, University of Medicine and Pharmacy of Craiova, 2 Petru Rareş Street, 200349 Craiova, Romania; 2Department of Anatomy, University of Medicine and Pharmacy of Craiova, 2 Petru Rareş Street, 200349 Craiova, Romania; ion.mindrila@umfcv.ro; 3Faculty of Pharmacy, University of Medicine and Pharmacy of Craiova, 2 Petru Rareş Street, 200349 Craiova, Romania; alice.buteica@umfcv.ro; 4Department of Biochemistry, University of Medicine and Pharmacy of Craiova, 2 Petru Rareş Street, 200349 Craiova, Romania; c_pisoschi@yahoo.com; 5Department of Rheumatology, University of Medicine and Pharmacy of Craiova, 2 Petru Rareş Street, 200349 Craiova, Romania; cristina.gofita@umfcv.ro; 6Department of Pharmacology, University of Medicine and Pharmacy of Craiova, 2 Petru Rareş Street, 200349 Craiova, Romania; iulia_paliu@yahoo.com

**Keywords:** nicotinamide riboside, monoiodoacetate-induced knee osteoarthritis, oxidative stress, nitrosative stress, inflammatory biomarkers, histology

## Abstract

Knee osteoarthritis (KOA), one of the most common orthopedic disorders concerning the adult population worldwide, is a condition characterized by progressive destruction of the articular cartilage and the presence of an inflammatory process. The aim of our study was to assess whether nicotinamide riboside (NR), a popular anti-aging supplement, can reduce the rate of cartilage destruction and alleviate the inflammatory response compared to the commonly prescribed collagen supplement in a murine monoiodoacetate (MIA)-induced KOA model. Twenty Wistar rats were randomly assigned to 4 groups: sham (S), MIA and NR, MIA and hydrolyzed collagen (HC), and MIA. At the end of the experiment, the right knees and blood samples were collected for histological assessment and biochemical evaluation of nitric oxide, malondialdehyde, total antioxidant capacity, reduced glutathione, glutathione peroxidase, superoxide dismutase, catalase, myeloperoxidase, and tumoral necrosis factor-alpha (TNF-α). The study determined that the treatment with NR in a similar dose with HC decreased blood/serum levels of oxidative stress biomarkers and the histological lesions in almost the same manner. The present findings suggest that NR may exhibit chondroprotective and anti-inflammatory effects in MIA-induced KOA in rats.

## 1. Introduction

Knee osteoarthritis is a progressive debilitating inflammatory joint condition caused by a cumulus of biochemical, molecular, and biomechanical imbalances that can be caused or aggravated by individual risk factors, such as smoking, obesity, certain physical activities, and genetic susceptibility [1,2]. Although the cardinal structural changes present in KOA are the disruption and loss of articular cartilage, involvement of the synovia, subchondral bone ligament menisci, and joint capsule is also present [3]. From a clinical standpoint, patients suffering from KOA exhibit pain, loss of function caused by the limitation of the range of movement, and abnormally elevated sensitivity to damp and/or cold environmental conditions or crepitation. As osteoarthritis progresses, symptomatology may increase in intensity [4].

KOA can be classified as primary, caused by age, genetic, or physical factors, and secondary, often the result of trauma, joint infection, or genetic or metabolic disorders. In primary osteoarthritis, age is a key contributor leading to mitochondrial dysfunction, which, in turn, results in elevated levels of oxidative stress, cell apoptosis, and the degradation of articular cartilage. It also involves inflammation and the dysfunction of chondrocytes, marked by an imbalance in the expression of matrix metalloenzymes and growth factors. Genetic factors also play an important role in KOA, factors associated with alteration of the inflammatory responses, and metabolic processes of the bone and cartilage. As for secondary OA, obesity is responsible for a higher level of inflammatory cytokines and adipokines produced by fat tissue. Moreover, obesity alters the normal mechanical stimulation of the joint, boosting matrix metalloproteinase (MMP) and interleukin 1 (IL-1) activity. Trauma can cause irreversible cartilage by inducing cell death and matrix degradation [5].

Articular cartilage changes or mechanical lesions can activate an inflammatory response characterized by an influx of monocytes, cells that are normally responsible for fibrosis and tissue repair, that later differentiate into macrophages, which can increase pro-inflammatory cytokine levels (interleukins IL-1β and IL-12 and TNF-α) and growth factors. In the early stages of KOA, CD4, and CD8 lymphocytes are increased in the synovium and produce cytokines and chemokines (IL-8, IL-17, TNF-α, MMP1, MMP3). Stromal cells and chondrocytes can also express a high level of pro-catabolic cytokines (IL-8, IL-12) [6,7]. Reactive oxygen species (ROS), proteases, and cytokines released by macrophages at a low level in the healthy synovium mediate inflammation and are critical for cell homeostasis and function. Physiological imbalances determine an overexpression of cytokines, such as IL-1β, IL-6, and TNF-α, and chemokines, which can lead to the oxidation of different macromolecules, a process that can alter cell function. High levels of oxidative stress increase the inflammatory response while altering glycosaminoglycans and collagen synthesis [7,8].

Although there is no cure or effective treatment for KOA, current therapeutic options aim to increase the quality of life by alleviating symptomatology and slowing the evolution of the disease.

KOA can be managed non-surgically in the initial stages of the disease, but more advanced stages require osteotomy, debridement, and arthroscopic lavage or joint replacement surgery [4,5]. Nonsurgical approaches involve pharmacological or nonpharmacological means, such as manual therapy, education, weight management, electrotherapy, ultrasound therapy, laser therapy [5], biomechanical interventions, and therapeutic exercise [9,10]. Pharmacological treatment of KOA implies the use of topical agents, such as non-steroidal anti-inflammatory drugs (NSAIDs), and orally administered pain relievers, such as classical NSAIDs, specific cyclooxygenase 2 inhibitors, or opioids. Invasive pharmacological means of treatment that involve intra-articular injections of hyaluronic acid, polymerized collagen, steroid drugs, and platelet-rich plasma have the advantage of directly delivering the therapeutic agent to the site of the inflammatory process [10,11,12]. Biomaterials, non-hydrogel polymers, such as poly N-isopropyl acrylamide and polyethylene glycol, hydrogels classified into proteins (collagen and gelatin), polysaccharides (alginates, heparin, chitosan, hyaluronic acid, dextran), and inorganic nanomaterials have been shown to improve the regeneratory function of the articular cartilage. Polysaccharide hydrogels have been shown to alleviate cartilage degeneration and pain in clinical studies, while protein-based hydrogels seem to reduce inflammation and inhibit osteoarthritis progression in murine models. Non-hydrogel polymers exhibit superior batch stability and mechanical properties. Inorganic nanoparticles reduce chondrocyte degeneration by loading and releasing drugs via mesoporous channels [5]. Moreover, curcumin nanoparticles show significant potential as an effective treatment for periprosthetic joint infections due to their antibacterial and anti-inflammatory properties. Exosome and growth factor therapy have great potential in treating osteoarthritis [5,13]. A novel strategy for managing knee osteoarthritis (KOA) involves the use of nutraceutical dietary supplements, which are compounds that hold promise in diminishing oxidative stress and exhibiting anti-inflammatory properties. Moreover, representatives of this class, such as hydrolyzed collagen, chondroitin sulfate, and glucosamine sulfate, reduce joint stiffness and facilitate cartilage regeneration [14]. Hydrolyzed collagen has been found to be effective in alleviating pro-inflammatory processes by modulating the activity of type 2 macrophage and T lymphocytes responsible for the secretion of anti-inflammatory mediators, such as IL-4, IL-10, and transforming growth factor-β (TGF-β), promoting thus extracellular matrix synthesis and cartilage repair [14,15]. Another widely used anti-aging nutraceutical food supplement, NR, a nicotinamide adenine dinucleotide (NAD^+^) precursor, is recommended for a wide array of diseases. NR has been shown to alleviate amyloid-β levels and certain inflammatory markers, such as IL-6, TNF-α, and IL-1. Regarding the musculoskeletal system, NR has been shown to reduce macrophage infiltration in damaged muscle tissue and to reduce the level of inflammatory cytokines in aged subjects [16,17]. Also, long-term NR supplementation exhibited cardioprotective effects by stimulating antioxidant gene expression [17].

NR has been shown to play a pivotal role in the interplay between hypoxia-inducible factor-2α (HIF-2α) and the nicotinamide phosphoribosyltransferase (NAMPT)-nicotinamide adenine dinucleotide-sirtuin axis (SIRT), contributing to osteoarthritis (OA) cartilage deterioration. It facilitates the activation of sirtuin family members by stimulating NAD^+^ synthesis, which, in turn, boosts HIF-2α protein stability and transcriptional activity. This dynamic interaction is essential for the expression of matrix-degrading enzymes and ensuing OA cartilage damage. Nicotinamide serves as a key mediator in the intricate mechanism linking HIF-2α and the NAMPT-NAD^+^ -SIRT axis in articular chondrocytes. Moreover, NR has been proven to regulate the differentiation of osteoblasts by enhancing the transcriptional activity of FOXO3A through the activation of Sirtuin 3, which translates into an increase in antioxidant enzyme expression [18,19].

The aim of our study was to determine whether NR could be a viable option in KOA compared with a commercially available HC supplement in an animal model of monoiodoacetate-induced osteoarthritis. Considering all the above, our focus was on antioxidant activity, myeloperoxidase (MPO), and TNF-α as inflammatory biomarkers.

## 2. Materials and Methods

### 2.1. Animals

Eight-week-old male Wistar rats (*n* = 20) weighing between 250 and 350 g were obtained from the Animal Facility Unit of the University of Medicine and Pharmacy of Craiova. All animals were housed individually in stainless steel ventilated cages, with water and standard laboratory chow ad libitum, under a constant room temperature of 20 °C and a 12-h light/dark cycle. All experimental procedures were approved by the Committee of Ethics and Scientific Deontology of the University of Medicine and Pharmacy of Craiova (No.57/16.02.2023). Prior to the start of the experimental procedures, the animals were acclimatized to the experimental environment for 24 h.

### 2.2. Study Design

The animals were randomly divided into 4 groups:

Sham (*n* = 5), non-osteoarthritic control rats that received an intraarticular injection with sterile saline;

NR (*n* = 5), animals that were injected with monoiodoacetate and received treatment with commercially available nicotinamide riboside supplements (300 mg/kg body weight) by oral gavage for 21 consecutive days;

HC (*n* = 5), animals that were injected with monoiodoacetate and received treatment with commercially available hydrolyzed collagen supplements (300 mg/kg body weight) by oral gavage for 21 consecutive days;

MIA (*n* = 5), animals that were injected with monoiodoacetate and did not receive any form of treatment.

### 2.3. Induction of Osteoarthritis

The animals were anesthetized using an intraperitoneal cocktail of ketamine (60–80 mg/kg)/xylazine (6–8 mg/kg) and the right knee was shaven with an electric trimmer and disinfected using an iodine solution. Except for the Sham group, which was injected with 50 µL sterile saline, all animals received an intraarticular injection of 2 mg/50 µL MIA solution using a standard 1 mL insulin syringe. A sterile saline solution was used as a vehicle for the MIA. After placing the knee in a 90° flexion, both MIA and saline solution were injected into the articular cavity through the infrapatellar ligament. The administration of NR and HC began one week after the injection of MIA (Merck^®^, Darmstadt, Germany, Sodium Iodoacetate 57858-5G-F). No other drugs or chemicals were administered to the animals before or after the start of the experimental procedure to avoid interfering with the results obtained.

### 2.4. Blood Sampling

After the animals were anesthetized using a ketamine (60–80 mg/kg) and xylazine (6–8 mg/kg) solution, blood samples were collected for cytokine and oxidative stress biomarkers through exsanguination via an intraventricular bleed in EDTA-coated containers. Following centrifugation at 1100× *g* for 10 min at 4 °C, the plasma and erythrocyte pellets were separated for further storage at −80 °C and analysis.

#### 2.4.1. Cytokine Assessment

TNF-α was assessed using a Rat ELISA kit (Biovendor^®^, Asheville, NC, USA, RAF-130R), following the manufacturer’s standard protocols. The results are expressed as pg/mL.

#### 2.4.2. Assessment of Oxidative and Nitrosative Stress Biomarkers

Nitric oxide (NO) was assessed using a standard Nitric Oxide Assay Kit (Sigma^®^, Setagaya City, Tokyo, Japan, MAK454-1KT). The results are expressed as µM.

MPO activity was assessed after centrifuging the plasma for 15 min at 1000× *g* at room temperature using a Rat Myeloperoxidase ELISA Kit (Abbexa^®^, abx513486). All procedures regarding the sample preparation followed the manufacturer’s standard protocols. The results are expressed as ng/mL.

For total antioxidant capacity (TAC), superoxide dismutase (SOD), glutathione peroxidase (GPx), catalase (CAT), reduced glutathione (GSH), and malondialdehyde (MDA) assessment, we used previously described methods [20,21].

Assessment of GPx activity was performed using a Ransel Kit (Randox Laboratories^®^, RS 504), following the standard protocol provided by the manufacturer. The results are expressed as U/L of hemolysate.

TAC was performed after the plasma samples were diluted in 0.01 M Phosphate Buffer Solution and 0.1 mM 2,2′-diphenyl-1-picrylhydrazyl radical reagent (DPPH) was added. After a 30-min incubation in a dark chamber at room temperature, the samples were centrifuged for 3 min at 20,000× *g* and 4 °C (Eppendorf 5417 R centrifuge), and the absorbance was read at 520 nm using a UV-VIS spectrophotometer Kruss. The results are reported as mmol DPPH/L.

SOD activity was assessed using a Ransod Kit (Randox Laboratories^®^, Parramatta, NSW, Australia, SD125) after centrifugation of the blood sample, multiple washes of the erythrocyte pellet with a sterile saline solution followed by lysing using cold redistilled water and treatment with Phosphate Buffer Solution (0.01 M, pH = 7). Afterward, hemolyzed samples were analyzed using the protocol provided by the manufacturer, and the absorbance was read at 505 nm with the same UV-VIS spectrophotometer. The results are expressed as SOD units/mL.

To assess the GSH level, the erythrocyte pellets were lysed using cold redistilled water and later centrifuged at 4020× *g* at 4 °C for 10 min. The lysate was treated with a 5% trichloroacetic acid solution, and the mixture was vortexed and then separated through centrifugation. After adding a mixture of 0.1 M Ellman’s Reagent and 0.07 M Phosphate Buffer Solution, the samples were incubated in a dark chamber and later analyzed at 412 nm using the same UV-VIS spectrophotometer. The final GSH concentration was calculated using a standard curve. The results are expressed as mg/dL.

For CAT assessment, the diluted hemolysate was added to a 0.07 M Phosphate Buffer Solution. Later, the obtained mixture was incubated for 10 min at 37 °C and treated with hydrogen peroxide. Samples were analyzed using a Beckman Coulter DU-65 UV-VIS spectrophotometer at 240 nm. The results are reported as units per mg of hemoglobin (U/mg Hb).

The assessment of MDA as a measure of lipid peroxidation began by adding one part plasma to nine parts of a 1:1 mixed solution of 35% trichloroacetic acid/0.2 M Tris-Cl and incubating the obtained mixture at room temperature for 10 min. Then, after adding a mixture of 2 M sodium sulfate in 0.05 M thiobarbituric acid, a second incubation at 95 °C for 45 min was performed, followed by cooling on an ice bed for 6 min. After adding a 70% trichloroacetic acid solution followed by vortexing and centrifuging, the super-natant was analyzed at 532 nm using a Kruss UV-VIS spectrophotometer. The results are expressed using the molar extinction coefficient of MDA-TBA abduct (1550 × 105 mM^−1^ cm^−1^).

### 2.5. Histopathological Analysis

After euthanasia of the rats, the entire right lower limbs were harvested and fixed for 72 h in a 10% formaldehyde solution. Later, the soft tissue was removed using a microtome razor. For the decalcification, the samples were placed on a shaker in a 5% formic acid solution, which was replaced daily for 13 days. The consistency of the sample was periodically assessed by poking it with a needle. Following the decalcification process, the excess bony tissue was removed, and the knees were cut through a frontal plane. After being placed in histological cassettes, the samples were processed for paraffin embedding using the laboratory standard protocol. The paraffin blocks were cut into 10 µm thick serial sections with a rotary Leica RM 2235 microtome and later collected on slides coated with poly-L-lysine. Subsequent to deparaffinization in successive xylene and alcohol baths, the samples were stained with Toluidine blue (Merk^®^, Toluidine Blue O T-3260) for visualizing the proteoglycan content and Hematoxylin/Eosin (Leica Biosystems^®^ Infinity 2.0) using standard protocols recommended by manufacturers and coverslipped using DPX mountant (Merck, DPX Mountant for histology, 44581). Light microscopy assessment of the samples was performed. The samples were blindly analyzed by two independent morphologists using a modified Mankin scale for scoring [22,23].

### 2.6. Statistical Analysis

Statistical analysis was performed using GraphPad Prism software version 8.0.1 (GraphPad Software^®^, San Diego, CA, USA). Continuous data are presented as the average ± standard deviation of the mean (SD). To determine the difference between the groups, we used *t*-test and one-way analysis of variance (ANOVA) with Mann–Whitney and Kruskal–Wallis tests for non-normally distributed data. A value of *p* < 0.05 was accepted as statistically significant.

## 3. Results

### 3.1. Cytokine and Oxidative and Nitrosative Stress Biomarkers

The effects of NR and HC on oxidative stress biomarkers and TNF-α levels are illustrated in Figure 1 and Table 1. Assessment of the antioxidant enzymes (CAT, GPx, and SOD) revealed decreased levels of activity in the MIA-induced KOA group compared to Sham rats. Our results showed that NR and HC treatments improved antioxidant enzyme activities. Both substances significantly increased the activity of CAT (*p* < 0.01) compared with the MIA group, while the other antioxidant enzymes investigated in this study did not reveal significantly different activities between NR, HC, and MIA groups. Reduced glutathione concentration, used to assess thiol levels, was significantly decreased in MIA-induced KOA rats compared to the Sham group (*p* < 0.01). HC treatment considerably increased the concentration of the non-enzymatic antioxidant GSH (*p* < 0.05), while the NR effect on GSH level was less significant. The analysis of MDA levels emphasized that lipid peroxidation in MIA-treated rats was higher than those of Sham rats (*p* < 0.01). Treatment of MIA rats with both HC and NR produced a significant decrease in MDA levels (*p* < 0.05), proving that they notably reduced oxidative stress compared to MIA animals.

MPO activity increased significantly in the MIA control group compared to Sham rats (*p* < 0.01). MPO activity was significantly decreased in HC- and NR-treated animals in comparison to the MIA group (*p* < 0.05), suggesting that NR and HC may play a role in decreasing oxidative stress pro-inflammatory reactions. Nitric oxide levels were greatly reduced in the HC group compared to the MIA and NR groups (*p* < 0.01), while TAC levels showed a statistically significant difference between the MIA and HC groups (*p* < 0.05). The levels of TNF-α were increased in the MIA-induced KOA group compared to those in the Sham group. The assessment of TNF-α presented statistically significant results only between the MIA- and HC-treated groups (*p* < 0.05), suggesting a stronger anti-inflammatory effect of HC compared to NR.

### 3.2. Histolopathological Analysis

The Mankin score was significantly higher in the MIA-treated group than in the Sham (*p* < 0.01), NR (*p* < 0.01), and HC groups (*p* < 0.05), as shown in Figure 2.

All animals included in this study, except those from the Sham group, exhibited an abnormal histological structure (Figure 3). Regarding the articular cartilage structure, the NR and HC groups presented lesions ranging from surface irregularities to clefts extending to the transitional zone, while the lesions in the MIA group were much more severe and characterized by loss of cartilage in the deep zone. Toluidine blue staining revealed that the NR group presented a lower glycosaminoglycan content in the superficial and middle zones of the cartilage. Animals from the HC group showed a lesser degree of injury, which extended to the upper transitional zone of the cartilage. Rats assigned to the MIA group presented a decreased glycosaminoglycan content in all three regions. The tidemark integrity was affected in all groups except for S. Apart from the criteria that compose the modified Mankin scale used in this study, we have observed meniscal lesions and inflammatory infiltrate concerning both the meniscus and the meniscofemoral and meniscotibial ligaments in the NR, HC, and MIA groups. An inflammatory infiltrate was also present in the synovia. Animals treated with HC or NR showed a lesser extent of lesions compared to the ones from the MIA group (Figure 3).

## 4. Discussion

The present study aimed to investigate whether NR could alleviate inflammation, morphological changes, and oxidative stress in an experimental model of MIA-induced KOA in rats, with MIA being an inhibitor of glyceraldehyde-3-phosphate dehydrogenase, which causes chondrocyte death and inflammation.

The MIA-induced KOA model, widely used in the literature, exhibits similar histopathological changes to human pathology, making it suitable for testing new therapeutic agents for osteoarthritis treatment [24,25].

Our histological findings are in accordance with the literature data regarding cartilage damage during osteoarthritis [26] and have shown that treatment with either NR or HC could alleviate the extent of lesions associated with KOA. Regarding articular cartilage structural changes, animals from the MIA group exhibited lesions that ranged from loss of cartilage in the superficial region to clefts extending into the deep zone and significant cartilage loss. The NR lot presented undulations of the articular surface, while the HC group changed from irregular surfaces to clefts that extended from the superficial to the middle zone. Animals assigned to the Sham work group showed minimal degenerative lesions. Proteoglycan content, assessed by toluidine blue staining, was severely decreased in all three cartilage zones of MIA rats, while NR and HC animals showed a decreased content in the superficial and middle zone. Regarding cellularity, both HC and NR presented similar lesions that varied from diffuse hypercellularity and clustering to hyper-cellularity. The HC group exhibited diffuse hypercellularity. As mentioned before, tidemark integrity was affected in all three workgroups. Animals from the NR group presented a highly statistically significant improvement (*p* < 0.01) compared to the MIA group, which suggests that NR may alleviate lesions associated with KOA.

The articular cartilage and its underlying bone are perpetually subjected to degradation and synthesis processes that necessitate the activation of catabolic and anabolic enzymes. While historically OA has been regarded as a non-inflammatory condition with structural damage being attributed only to mechanical stress, studies have shown that inflammation plays a crucial role in disease development [27]. Synovial inflammation is present in the early stages of the disease, and it is associated with the magnitude of the structural changes. The activation of the inflammatory response can be a consequence of mechanical injury. Later on, an influx of monocytes and macrophages results in an increase of cytokines (IL-6, IL-1β, TGF-β, and TNF-α), chemokines (IL-17-induced CCL2, CCL20, CCL3, CCL4), and pro-inflammatory enzymes (aggrecanase, collagenase, MMPs, MPO). Moreover, chondrocytes themselves promote cartilage degradation by means of autocrine and paracrine activation, which causes an overexpression of catabolic cytokines. Consecutively, activated monocytes and circulating cytokines, such as TNF-α and IL-6, maintain the inflammatory process and promote structural changes [6].

Recent studies have suggested that excessive generation and storage of reactive oxy-gen species (ROS), defined as free radicals or molecules that contain oxygen, such as hydrogen peroxide, hydroxyl radicals, superoxide anion radicals, peroxynitrite anions, and hypochlorite ions, is the main cause of osteoarthritis. It is important to mention that inappropriate mechanical stimulation leads to ROS overproduction, which can cause cell death or depolymerization of the hyaluronic acid. ROS are generated mainly at the mitochondrial level as a secondary product of normal biological processes. Moreover, chondrocytes produce low levels of ROS as part of an intracellular signaling pathway to modulate apoptosis, cartilage homeostasis, cytokine production, and extracellular matrix synthesis. In KOA, elevated levels of ROS determine cellular membrane damage and alteration of the nucleic acids in chondrocytes but also matrix degradation by decreasing proteoglycan and collagen synthesis. ROS-induced oxidative stress has been defined as an imbalance between ROS production and antioxidant defense mechanisms, such as peroxiredoxins (PRDXs), SOD, CAT, and GPx [28,29,30].

Catalase is an enzyme with an antioxidant capacity that is responsible for the catalysis of hydrogen peroxide decomposition in oxygen and water. By alleviating ROS levels, CAT improves the chondrocyte’s survival rate, thus reducing the severity of the lesions [31]. Recent studies indicate that CAT regulates processes such as apoptosis, proliferation, transduction, and differentiation, in which the second messenger implied is hydrogen peroxide [30]. Although other literature studies concerning NR showed an improvement in CAT levels in cell cultures and animal models, to our knowledge, there are no current records concerning CAT in osteoarthritis [31,32].

One of the key findings of our study was that both NR and HC significantly increased CAT activity in the same manner (*p* < 0.01).

Both GPX and SOD, components of the endogenous antioxidant system, are involved in neutralizing free radicals. SOD, an enzyme associated with low levels of synovial fluid in KOA when decreased, and GPx, another element involved in reducing hydroperoxide radical, [33,34,35], were not found to be significant either in the NR or in the HC groups, the only statistically significant difference being between the MIA-treated and Sham groups.

MPO is a member of the peroxidase subfamily, mainly expressed by leukocytes, lymphocytes, and neutrophils, and is responsible for generating ROS in response to microbial aggression. An uncontrolled release of MPO increases inflammatory events and can also cause damage to healthy structures, even without the presence of an infectious factor. MPO is incriminated in several major pathologies, such as liver disease, rheumatoid arthritis, cancer, and diabetes [36].

MDA, a marker of lipid peroxidation, exhibits increased levels in KOA, which is responsible for the oxidation and degradation of collagen. Comparative studies of free radical types in the synovial fluid and plasma of patients suffering from KOA have shown elevated values of MDA both in plasma and the synovial fluid [33]. In the present study, MDA levels were elevated in the MIA groups, indicating increased lipid peroxidation and cellular membrane damage. Both NR and HC treatment reduced statistically significant MDA and MPO levels in a similar manner (*p* < 0.05).

NO plays an important role in both physiological and pathological processes. In KOA, excessive production of NO maintains a high level of inflammatory cytokines and pro-motes matrix degradation by altering the balance between degradation and synthesis of collagen and proteoglycan in the cartilage [37].

TNF-α has an important role in the evolution of KOA. A literature study has shown that an increase of TNF-α along with interleukin-1β impaired the articular chondrocyte function [38].

The results of our study are in agreement with Hamza et al. [39], who demonstrated that TNF-α and inflammatory interleukin levels were elevated in OA knees as compared to MIA controls. The present study has shown that only HC treatment significantly reduced the NO and TNF-α levels (*p* < 0.05), while for NR, although there is a tendency for the result to be notable, the decrease is not statistically significant.

A decreased level of TAC, another endogenous tool for neutralizing ROS, was reported for patients with OA [40]. In our study, HC was found to be effective in increasing the TAC level, while NR showed only a modest improvement but without statistical significance.

Considering the effects observed in the HC group, our results are in accordance with the literature reports, as we observed an amelioration of both oxidative and inflammatory biomarkers [41].

The use of nicotinamide riboside as an adjuvant therapy in knee osteoarthritis holds promise due to its potential to enhance energy production, reduce inflammation, support chondrocyte function, and counteract oxidative stress [42]. While further research is needed to confirm its efficacy, NR supplementation represents an intriguing avenue for improving the management of knee osteoarthritis and potentially enhancing the quality of life of individuals affected by this condition. Moreover, both supplements used in the study were available on the pharmaceutical market in accordance with the national law applied by our country. Concerning the international opinion on these supplements, NR is considered the safest form among the NAD^+^ precursor supplements by three international authorities: Food and Drug Administration (FDA), the Therapeutic Goods Administration (TGA) of Australia, and Health Canada (HC) [40]. As for HC, both the European Commission for Health and Consumer Protection and the World Health Organization have classified HC supplements as safe, while the Food and Drugs Administration (FDA) concerned the ingredient used for the preparations of collagen peptides (gelatin) with a favorable report [43].

Regarding the safety assessment, NR was not found to be cytotoxic or mutagenic at doses up to 2000 mg/kg. Also, the toxicological studies concerning the adverse effects in line with the administered dose have classified the dose of 300 mg/kg, used in our study, as a safe and side effects-free option [44].

Furthermore, in regard to our registered results, another basis that justifies the choice of NR as an optimal nutraceutical supplement among other NAD^+^ precursors is represented by its increased bioavailability and the greater potential to elevate the NAD^+^ levels [17]. It is worth mentioning that the present study, which has demonstrated a potential beneficial effect of NR in a murine model of MIA-induced KOA, is a preliminary one. The limitations of this article are represented by the lack of far larger animal work groups, which could reveal whether the actual improvement trend regarding certain biomarkers could become statistically significant. Other limitations are constituted by a lack of more experimental means such as immunochemistry, immunofluorescence, and Western blot analysis, which will be addressed in the upcoming studies.

## 5. Conclusions

Regarding HC supplementation, the results of our experiment are in accordance with the existing literature data. The present study has demonstrated that NR, a precursor of NAD^+^ can alleviate certain oxidative stress markers and histological lesions associated with KOA, making it a highly potential treatment option in the early stages of the disease. It is essential to consider NR treatment as a complementary approach rather than a standalone solution. While these early findings are encouraging, it is important to acknowledge that more research is needed to fully understand the efficacy and safety of NR treatment for knee osteoarthritis. Further clinical trials with larger sample sizes and longer durations are required to confirm these preliminary results and to establish optimal dosages and treatment regimens.

## Figures and Tables

**Figure 1 jcm-12-06920-f001:**
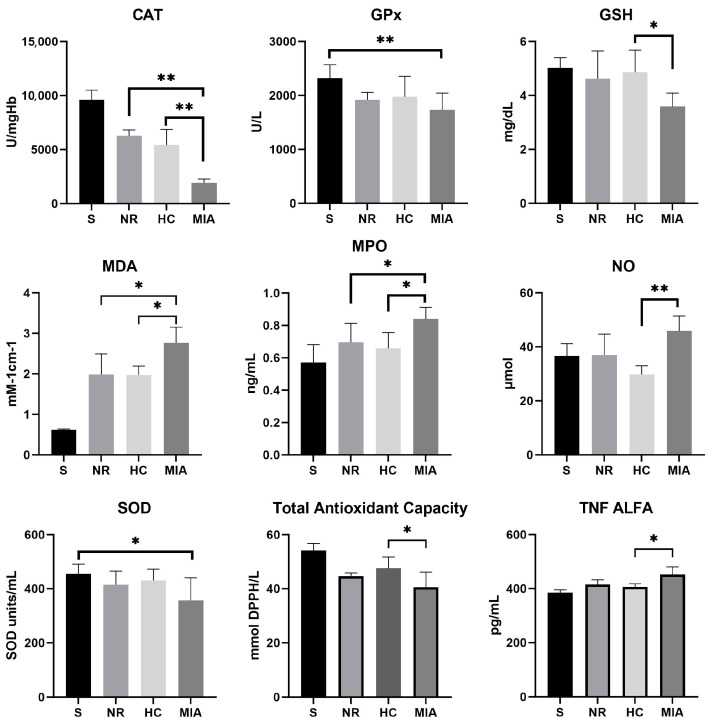
Variation and statistical analysis of cytokine and oxidative stress biomarkers (*, *p* < 0.05; **, *p* < 0.01).

**Figure 2 jcm-12-06920-f002:**
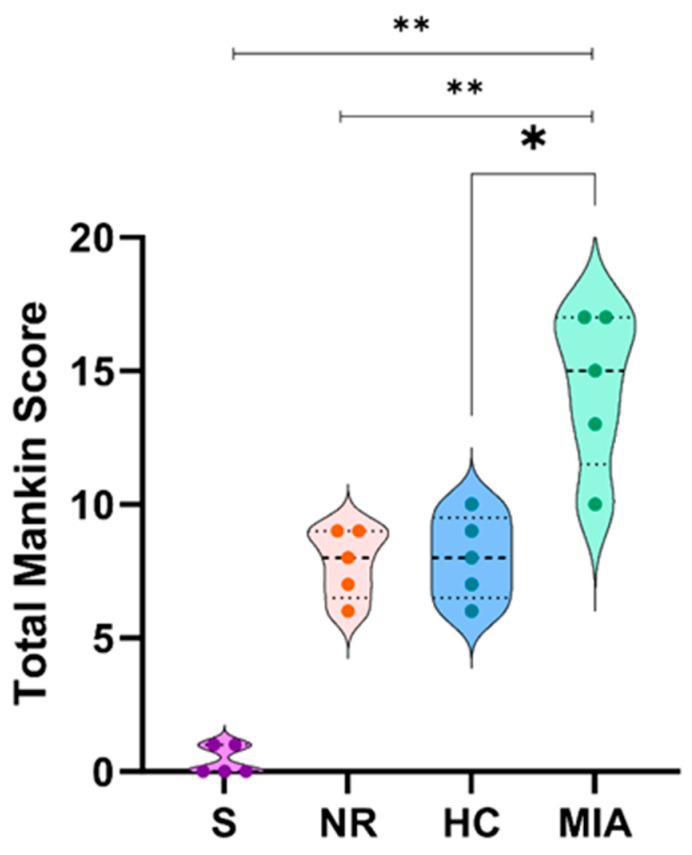
The Mankin scoring of the studied groups. The Mann–Whitney test was used to compare the Mankin score between the groups. *, *p* < 0.05; **, *p* < 0.01.

**Figure 3 jcm-12-06920-f003:**
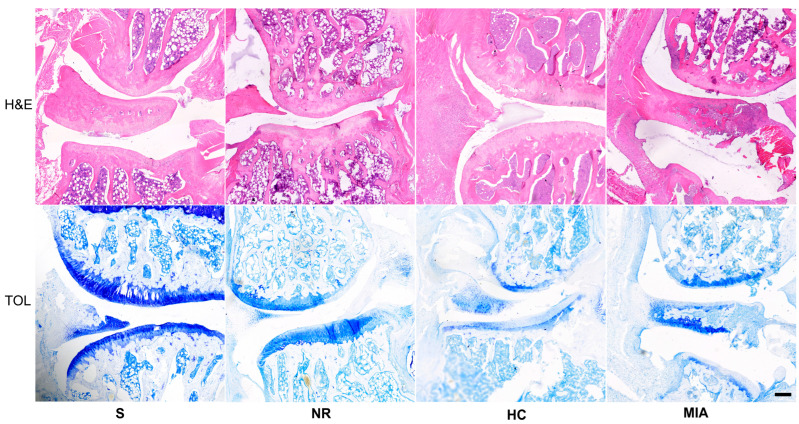
Histological aspects of the articular lesions in the studied lots. H&E and toluidine blue (TOL) staining. Bar = 250 µm.

**Table 1 jcm-12-06920-t001:** Mean values and ± standard deviation of biochemical markers (S.D. = standard deviation).

	S		NR		HC		MIA	
	Mean	S.D.	Mean	S.D.	Mean	S.D.	Mean	S.D.
CAT [U/mg Hb]	9618.00	888.40	6286.00	536.30	5425.00	1445.00	1940.00	333.50
GPx [U/L]	2319.00	250.50	1922.00	134.30	1975.00	377.70	1736.00	307.00
GSH [mg/dL]	5.02	0.38	4.62	1.02	4.86	0.81	3.59	0.50
MDA [mM^−1^cm^−1^]	0.62	0.02	1.98	0.51	1.97	0.22	2.76	0.39
MPO [ng/mL]	0.57	0.11	0.70	0.12	0.66	0.10	0.84	0.07
NO [μmol]	36.64	4.54	36.96	7.76	29.76	3.29	45.92	5.57
SOD [units/mL]	454.50	36.36	414.60	50.92	430.60	41.71	356.60	84.21
TAC [mmol DPPH/L]	54.18	2.06	44.66	0.99	47.60	3.37	40.50	4.58
TNF Alfa [pg/mL]	385.20	10.26	414.80	17.58	406.80	11.10	452.20	28.52

## Data Availability

All data are present in the main text.

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
