# Peer review of "The Potential Benefic Effect of Nicotinamide Riboside in Treating a Murine Model of Monoiodoacetate-Induced Knee Osteoarthritis"

_jcm, 2023, doi:10.3390/jcm12216920_

Round 1

Reviewer 1 Report

Comments and Suggestions for Authors

The authors evaluated whether nicotinamide nucleoside could reduce the cartilage destruction rate and inflammatory response of mouse knee osteoarthritis induced by monoiodoacetate (MIA), and compared with hydrolyzed collagen. The experimental results showed that nicotinamide nucleoside can alleviate certain oxidative stress markers and histological lesions associated with KOA. Overall, nicotinamide nucleoside has excellent potential for the treatment of KOAbut will still require major modifications.

1.    The authors should strengthen the manuscript writing and improve the readability of this article.

2.       In the Introduction section, the authors do not provide sufficient information about KOA. I advised that authors could add the Pathogenesis of KOA.

3. The cytotoxicity test of nicotinamide riboside is essential, and the authors are recommended to improve the relevant content.

4. The explanation about medication methods are too simple. More details may be found inLei Y, Zhang Q, Kuang G, et al. Functional biomaterials for osteoarthritis treatment: from research to application [J]. Smart Medicine, 2022, 1(1): e20220014.” and “Lin Z, Jiang S, Ye X, et al. Antimicrobial curcumin nanoparticles downregulate joint inflammation and improve osteoarthritis [J]. Macromolecular Research, 2023: 1-9. ”.

5. The authors did not perform enough experiments to conduct the effects of nicotinamide riboside. It is recommended that the author add more cell experiments, such as immunofluorescence and Western blot experiments.

Comments on the Quality of English Language

 Minor editing of English language required

Author Response

Dear Reviewer,

We thank you for your time and effort in reviewing our manuscript. The feedback has been invaluable in improving the content and presentation of the paper.

We have revised our manuscript according to your comments. The changes are highlighted in yellow in the attached manuscript, and our point-by-point responses are given below:

  1. The authors should strengthen the manuscript writing and improve the readability of this article.

Response: 

By adding the suggested topics, we hopefully improved the readability of this article.

  1. In the Introduction section, the authors do not provide sufficient information about KOA. I advised that authors could add the Pathogenesis of KOA.

Response: 

We have addressed the presented issue by adding a paragraph about the pathogenesis of KOA as you requested:

“KOA can be classified as primary caused by age, genetic or physical factors, and secondary, often the result of trauma, joint infection, genetic or metabolic disorders. In primary osteoarthritis, age is a key contributor, leading to mitochondrial dysfunction, which, in turn, results in elevated levels of oxidative stress, cell apoptosis, and the degradation of articular cartilage. It also involves inflammation and the dysfunction of chondrocytes, marked by an imbalance in the expression of matrix metalloenzymes and growth factors. Genetic factors also play an important role in the KOA, factors associated with alteration of the inflammatory responses, and metabolic processes of the bone and cartilage. As for the secondary OA, obesity is responsible for a higher level of inflammatory cytokines and adipokines produced by the fat tissue. Moreover, obesity alters the normal mechanical stimulation of the joint, boosting matrix metalloproteinase (MMP) and interleukin 1 (IL-1) activity. Trauma can cause irreversible cartilage by inducing cell death and matrix degradation [5].”

  1. The cytotoxicity test of nicotinamide riboside is essential, and the authors are recommended to improve the relevant content.

Response: 

Considering that the cytotoxicity test for nicotinamide riboside was already performed by other authors, we adapted our manuscript with the relevant information accordingly:

“Regarding the safety assessment, NR was not found neither cytotoxic nor mutagenic at doses up to 2000 mg/kg. Also, the toxicological studies concerning the adverse effects in line with the administered dose, have classified the dose of 300 mg/kg, used in our study, as a safe and side effects-free option [44].” 

  1. The explanation about medication methods are too simple. More details may be found in “Lei Y, Zhang Q, Kuang G, et al. Functional biomaterials for osteoarthritis treatment: from research to application [J]. Smart Medicine, 2022, 1(1): e20220014.” and “Lin Z, Jiang S, Ye X, et al. Antimicrobial curcumin nanoparticles downregulate joint inflammation and improve osteoarthritis [J]. Macromolecular Research, 2023: 1-9. ”.

Response: 

We have addressed the presented issue by adding relevant information regarding physical therapy, and surgical and biotechnological means of managing osteoarthritis using the suggested material. The provided content improves the quality of the Introduction of this hereby study. 

“KOA can be managed non-surgically in the initial stages of the disease, but more advanced stages require osteotomy, debridement and arthroscopic lavage or joint replacement surgery  [4,5]. Nonsurgical approaches involve pharmacological or nonpharmacological means, such as manual therapy, education, weight management, electrotherapy, ultrasound therapy, laser therapy [5], biomechanical interventions, and therapeutic exercise [8, 9]. Pharmacological treatment of KOA implies the use of topical agents, such as non-steroidal anti-inflammatory drugs (NSAIDs), and orally administered pain relievers either classical NSAIDs, specific cyclooxygenase 2 inhibitors, or opioids. Invasive phar-macological means of treatment that involve intra-articular injections of hyaluronic acid, polymerized collagen, steroid drugs, and platelet-rich plasma have the advantage of directly delivering the therapeutic agent to the site of the inflammatory process [9–11]. Biomaterials, non-hydrogel polymers such as poly N‐isopropyl acrylamide or polyethylene glycol, hydrogels classified into proteins (collagen and gelatin), and polysaccharides (alginates, heparin, chitosan, hyaluronic acid, dextran), and inorganic nanomaterials have been shown to improve the regeneratory function of the articular cartilage. Polysaccharides hydrogels have been shown to aleviate cartilage degeneration and pain in clinical studies while protein based hydrogels seem to reduce inflamation and to inhibit osteoarthritis progresion in murine models. Non-hydrogel polymers exhibit superior batch stability and mechanical propieties. Inorganic nanoparticles reduce condrocyte degeneration by loading and releasing drugs via mesoporous channels [5]. Moreover, curcumin nanoparticles show significant potential as an effective treatment for periprosthetic joint infections due to their antibacterial and anti-inflammatory properties. Exosome and growth factor therapy hold a great potential in treating osteoarthritis [5,13].

  1. The authors did not perform enough experiments to conduct the effects of nicotinamide riboside. It is recommended that the author add more cell experiments, such as immunofluorescence and Western blot experiments.

Response:

The hereby study is a preliminary one and was performed to assess whether NR could improve the outcomes of certain biomarkers and histological lesions in a murine model MIA-induced KOA. Thank you for pointing out the current limitations of this article, facts that we have mentioned in the Discussion section,  and further investigation possibilities. We will address the suggested analyses in an upcoming work, but at this time we do not have the means to perform such experiments.

“It is worth mentioning that the hereby study, which has demonstrated a potential beneficial effect of NR in a murine model of MIA-induced KOA,  is a preliminary one. The limitations of this article are represented by the lack of far larger animal work groups, which could reveal if the actual improvement trend regarding certain biomarkers could become statistically significant. Other limitations are constituted lack of more experimental means such as immunochemistry, immunofluorescence, and Western blot analysis which will be addressed in the upcoming studies.”

Reviewer 2 Report

Comments and Suggestions for Authors

The authors of the manuscript, ‘The potential benefic effect of nicotinamide riboside in treating a murine model of monoiodoacetate-induced knee osteoarthritis’ have tried to investigate the role and relevance of nicotinamide riboside, a popular anti-aging supplement, which can reduce the rate of cartilage destruction and alleviate the inflammatory response compared to the commonly prescribed collagen supplement in a murine monoiodoacetate (MIA) induced KOA model.      The paper has been written nicely with appropriate methodology but it will be better if some following points are taken care of.

1.         Write the complete word with the abbreviation in the parenthesis, where it comes for the first time in the text, such as NR in line 86, MPO in line 97. Give their full form there. Check this for the whole text.

2.         The relationship between nicotinamide riboside (NR) and knee osteoarthritis (KOA) has been investigated lately (PMIDs: 37464093, 26209889), so authors may write a line or two regarding the gap of knowledge in the introduction section.

3.         Authors must write a few lines regarding the status of commercially used NR and HC supplements that whether they were FDA-approved or unapproved products.

4.         Authors must give statistical values of the parameters in a table endorsing Figure 1 data for clarity.

5.         Authors should write a few lines regarding the 'limitations of the study’

Author Response

Dear Reviewer,

We thank you for your time and effort in reviewing our manuscript. The feedback has been invaluable in improving the content and presentation of the paper.

We have revised our manuscript according to your comments. The changes are highlighted in yellow in the attached manuscript, and our point-by-point responses are given below:

  1. Write the complete word with the abbreviation in the parenthesis, where it comes for the first time in the text, such as NR in line 86, MPO in line 97. Give their full form there. Check this for the whole text.

Response:

We have addressed this issue regarding word abbreviation.

  1. The relationship between nicotinamide riboside (NR) and knee osteoarthritis (KOA) has been investigated lately (PMIDs: 37464093, 26209889), so authors may write a line or two regarding the gap of knowledge in the introduction section.

Response: 

We have addressed the issue regarding the gap of knowledge regarding the relationship between nicotinamide riboside and knee osteoarthritis Introduction section of our study.

“NR has been shown to play pivotal role in the interplay between hypoxia-inducible factor-2α (HIF-2α) and the nicotinamide phosphoribosyltransferase (NAMPT)-nicotinamide adenine dinucleotide-sirtuin axis (SIRT), contributing to osteoarthritis (OA) cartilage deterioration. It facilitates the activation of sirtunin family members by stimulating NAD+ synthesis, which, in turn, boosts HIF-2α protein stability and transcriptional activity. This dynamic interaction is essential for the expression of matrix-degrading enzymes and the ensuing OA cartilage damage. Nicotinamide serves as a key mediator in the intricate mechanism linking HIF-2α and the NAMPT-NAD+ -SIRT axis in articular chondrocytes. Moroever, NR has been proven to regulate the differentiation of osteoblasts by enhancing the transcriptional activity of FOXO3A through the activation of Sirtuin 3 which translates in an increase of antioxidant enzyme expression [18,19]. The aim of our study was to assess whether NR could be a viable option in KOA in comparison with a commercially available HC supplement in an animal model of monoiodoacetate-induced osteoarthritis. Considering all the above, our focus was on the antioxidant activity and MPO and TNF-α as inflammatory biomarkers.”

  1. Authors must write a few lines regarding the status of commercially used NR and HC supplements that whether they were FDA-approved or unapproved products.

Response: 

In our country, the main regulator for the approval of pharmaceutical supplements is the Romanian Ministry of Health based on law no.56/31.03.2021. However, owing to your suggestion, we verified the international standards and we concluded the following statements that were added to the text:

“The supplements used in the study were available on the pharmaceutical market in accordance with the national law applied by our country. Concerning the international opinion on these supplements, NR is considered the safest form among the NAD+ precursors supplements by three international authorities: Food and Drugs Administration (FDA), the Therapeutic Goods Administration (TGA) of Australia, and by the Health Canada (HC) [X]. As for the HC, both the European Commission for Health and Consumer Protection and the World Health Organization have classified the HC supplements as safe, while the Food and Drugs Administration (FDA) concerned the ingredient used for the preparations of collagen peptides (gelatin) with a favorable report [43].

Regarding the safety assessment, NR was not found to be cytotoxic nor mutagenic at doses up to 2000 mg/kg. Also, the toxicological studies concerning the adverse effects in line with the administered dose, have classified the dose of 300 mg/kg, used in our study, as a safe and side effects-free option [44].”

  1. Authors must give statistical values of the parameters in a table endorsing Figure 1 data for clarity.

Response: 

We have added a table named “Table 1.  Mean values and ± Standard deviation of biochemical markers (S.D=standard deviation)” with the statistical values of the parameters from Figure 1.

  1. Authors should write a few lines regarding the 'limitations of the study’

Response: 

Thank you for pointing out this important aspect. We have addressed this issue in the Discussion topic of the article.

“It is worth mentioning that the hereby study, which has demonstrated a potential beneficial effect of NR in a murine model of MIA-induced KOA,  is a preliminary one. The limitations of this article are represented by the lack of far larger animal work groups, which could reveal if the actual improvement trend regarding certain biomarkers could become statistically significant. Other limitations are constituted lack of more experimental means such as immunochemistry, immunofluorescence, and Western blot analysis which will be addressed in the upcoming studies.”

Round 2

Reviewer 1 Report

Comments and Suggestions for Authors

This revised manuscript has addressed most of my previous comments, so l am pleased to recommend it for publication as it is.